

# Daytime Isoprene Nitrates Under Changing NO$_x$ and O$_3$

Alfred W. Mayhew[1], Peter M. Edwards[1], Jaqueline F. Hamilton[1,2]

[1] Wolfson Atmospheric Chemistry Laboratories, Department of Chemistry, University of York, Heslington, York, UK
[2] National Centre for Atmospheric Science, University of York, York, UK

*Correspondence to*: Jaqueline F. Hamilton (jacqui.hamilton@york.ac.uk)

**Abstract.** Organonitrates are important species in the atmosphere due to their impacts on NO$_x$, HO$_x$, and O$_3$ budgets, and their potential to contribute to secondary organic aerosol (SOA) mass. This work presents a steady-state modelling approach to assess the impacts of changes in NO$_x$ and O$_3$ concentrations on the organonitrates produced from isoprene oxidation. The diverse formation pathways to isoprene organonitrates dictate the responses of different groups of organonitrates to changes

in O$_3$ and NO$_x$. For example, organonitrates predominantly formed from the OH-initiated oxidation of isoprene favour formation under lower ozone and moderate NO$_x$ concentrations, whereas organonitrates formed via day-time NO$_3$ oxidation show the highest formation under high O$_3$ concentrations with little dependence on NO$_x$ concentrations. Investigating the response of total organonitrates reveals complex and non-linear behaviour with implications that could inform expectations of changes to organonitrate concentrations as efforts are made to reduce NO$_x$ and O$_3$ concentrations, including a region of

NO$_x$-O$_3$ space where total organonitrate concentration is relatively insensitive to changes in NO$_x$ and O$_3$. These conclusions are further contextualised by estimating the volatility of the isoprene organonitrates revealing the potential for high concentrations of low volatility species under high ozone conditions.

## 1    Introduction

Organonitrates are important species in the atmosphere due to their potential to impact on NO$_x$, HO$_x$, and O$_3$ budgets through

gas-phase chemistry.(Emmerson and Evans, 2009; Bates and Jacob, 2019; Schwantes et al., 2019; Schwantes et al., 2020; Vasquez et al., 2020) Their relatively low volatility also results in the potential to form secondary organic aerosol (SOA) via condensation onto existing particles, and some organonitrates can undergo reactive uptake to the particle phase.(Hallquist et al., 2009; Schwantes et al., 2019; Palmer et al., 2022) Isoprene organonitrates have been widely studied due to the large emissions of isoprene resulting in the relevance of isoprene chemistry to a range of environments around the globe.

(Guenther et al., 2006; Pye et al., 2015; Reeves et al., 2021; Tsiligiannis et al., 2022)

Isoprene hydroxynitrate (IHN) is widely studied due to its formation from OH oxidation in the presence of NO resulting in high concentrations during the daytime (Figure 1).(Xiong et al., 2015; Wennberg et al., 2018) IHN also has formation routes from oxidation with the nitrate radical (NO$_3$). Other commonly studied isoprene mononitrates include isoprene carbonyl nitrate (ICN) and isoprene hydroperoxy nitrate (IPN). IPN forms through the initial NO$_3$ oxidation of isoprene to form an

isoprene nitrooxyperoxy radical (INO$_2$). Reaction of INO$_2$ with HO$_2$ then forms IPN. ICN has a range of formation pathways initiated by OH and NO$_3$ oxidation. Isoprene dinitrate (IDN) can also form from INO$_2$ by its reaction with NO.



Isoprene epoxides, such as isoprene epoxydiols (IEPOX), have long been of interest due to their potential to contribute to SOA by reactive uptake to acidified particles. (Paulot et al., 2009; Surratt et al., 2010) Later work outlined the similar SOA-forming properties for the nitrated epoxide, isoprene nitrooxyhydroxy epoxide (INHE), with the first proposed formation route to INHE involving the OH oxidation of IPN (Figure 2).(Schwantes et al., 2015) Recent work aiming to improve the representation of isoprene $NO_3$ chemistry in chemical mechanisms highlighted a previously unrepresented reaction pathway to forming nitrated epoxides from alkoxy radicals (RO). (Vereecken et al., 2021; Carlsson et al., 2022) This alkoxy-epoxidation pathway provides an alternative formation route to INHE that doesn't rely on a stable intermediate or the presence of OH (Figure 3). Additionally, three more nitrated epoxides can result from this pathway: isoprene nitrooxycarbonyl epoxide (INCE), isoprene nitrooxyhydroperoxy epoxide (INPE), and isoprene dinitrooxy epoxide (IDNE).

The motivation for the work presented here stems from findings from the 2017 Atmospheric Pollution and Human Health in a Chinese Megacity (APHH) summer campaign in Beijing, showing the role of $NO_3$ in the formation of isoprene organonitrates and their successive particle-phase products, including in the afternoon due to the presence of high $O_3$ concentrations. (Hamilton et al., 2021; Newland et al., 2021) The presence of high $O_3$ concentrations increased the conversion of NO to $NO_2$ which subsequently reduced the loss of $NO_3$ to reaction with NO. Geyer *et al.* first highlighted this in 2003, calculating daytime $NO_3$ mixing ratios of up to 2-5ppt during the afternoon during a haze period in La Porte, Texas.(Geyer, 2003) Since then, daytime $NO_3$ has been highlighted as a potentially important chemical pathway from a range of field campaigns in various cities around the world. (Brown et al., 2005; Osthoff et al., 2006; Khan et al., 2015; Xue et al., 2016; Wang et al., 2020; Foulds et al., 2021) Daytime $NO_3$ chemistry has also been shown to be potentially significant under forest canopies, where photolytic processes are diminished. (Forkel et al., 2006; Hu et al., 2013; Mermet et al., 2021)

The co-occurrence of organonitrate formation from OH and $NO_3$ chemistry, along with the multi-stage chemistry often required for their formation, results in the potential for complex dependencies on $NO_x$ and $O_3$ concentrations. This work describes efforts to investigate the dependence of daytime isoprene organonitrate concentrations on changes to $NO_x$ and $O_3$ through a series of steady-state models. While the data presented here solely focusses on nitrated species resulting from isoprene, similar variations in organonitrate speciation under different $NO_x$-$O_3$ regimes are likely to hold for any VOC which can undergo oxidation by both OH and $NO_3$ radicals.

## 2    Experimental

### 2.1    Model Description

The goal of this work is to investigate changes in $NO_x$ and $O_3$ on the chemistry of isoprene nitrates in the afternoon period in Beijing. To do this, the models should demonstrate the favoured reaction pathways under different oxidant concentrations in the absence of other variables. This means that physical and photolytic processes should be held constant, i.e. the models will describe the chemistry occurring at a representative point in the day. Species must also be allowed to reach their steady-state concentrations in order to eliminate the role of the model spin-up period on resultant species concentrations. Comparison of the concentrations of species in these so-called steady-state models then allows for conclusions to be drawn as to the preferred oxidation products under various conditions.



All models described in this work were run using AtChem2, an open-source zero-dimensional box model. (Sommariva et al., 2019) All models also made use of the Isoprene mechanism published by Vereecken *at al.* (henceforth the FZJ Mechanism) which aimed to improve the representation of NO$_3$ chemistry of isoprene by building on chemistry from the Master Chemical Mechanism (MCM) and the review of isoprene chemistry published by Wennberg *et al.* (Jenkin et al.,

2015; Wennberg et al., 2018; Vereecken et al., 2021; Carlsson et al., 2022)

The steady-state models are sets of models run at a range of fixed NO$_x$ and O$_3$ mixing ratios. Models were run for NOx mixing ratios up to 45 ppb and O$_3$ mixing ratios of 140 ppb, corresponding to the upper limit of measurements made in the Beijing 2017 campaign. In order to provide additional OH reactivity, a constant concentration of methane was added to all of the models to ensure that the modelled OH reactivity matched measured values under Beijing-like conditions, this is

discussed further in the Model Validation section. The required mixing ratio corresponded to 82 ppm of methane in all of the models. The modelled concentration of species is taken as the final concentration after 5 model days, after all species had been allowed to reach steady-state concentrations. Each model was run at photolysis conditions corresponding to those calculated by AtChem2 for 16:00 local time in Beijing, China. This was around the time of peak daytime NO$_3$ concentrations in Beijing, before concentrations rapidly increased during sunset. To ensure steady-state was reached in a reasonable time,

and to provide loss routes for species without losses, species were removed from the model at a dilution rate of $2.31 \times 10^{-5}$ s$^{-1}$, corresponding to a dilution lifetime of 12 hours.

All models were run at a temperature of 298.15 K, a pressure of 1013 mbar, and a relative humidity of 50%. The latitude, longitude, and date used for photolysis calculations were 39.909°, 116.398°, and 2022-06-01. NO$_x$ was constrained by adjusting the NO and NO$_2$ concentrations at the beginning of each time step such that the total NO$_x$ matched the desired

concentration but the ratio of NO and NO$_2$ remained constant.

Models were also run to simulate conditions in the Amazon region, with VOC concentrations adjusted to match observations of isoprene concentrations and OH reactivity in this region. These models were run at a higher isoprene concentration of 5 ppb, and methane concentrations of 100 ppm. (Williams et al., 2016; Pfannerstill et al., 2021; Langford et al., 2022) The latitude and longitude values used corresponded to the city of Manaus and were -3.132° and -60.01° respectively. The time

of day was kept at 16:00 local time.

These models are designed for comparison between one another to gain insight into the impact of changes in NO$_x$ and O$_3$ on organonitrate concentration. The models show the concentrations at steady-state for the provided photolysis conditions, chemistry, and dilution rate, which is in contrast to the constantly changing photolysis and dilution encountered under ambient conditions. However, Section 3.1 illustrates that the conclusions made in this paper are applicable to the Beijing

afternoon conditions being investigated and that the conclusions are robust to changes in the modelling approach.

**2.2   Model Isopleths**

Throughout this paper, the model results are investigated through the use of isopleth plots. These plots consist of the steady-state concentration of a species (or another model output such as OH reactivity) in each of the models plotted as a coloured circle at the corresponding position on a set of NO$_x$-O$_3$ axes. The colour scale is indicated by a colour bar placed alongside



each set of axes and will be a different scale for each plot. A continuous colour gradient is then overlaid on the axes by interpolation over a triangular grid of the model points. 10 contour lines are also drawn over the top of each plot to highlight the contour shape. These lines are equally spaced in the coloured dimension (e.g. species concentration), meaning close vertical lines would correspond to a strong sensitivity to changes in $O_3$ and close horizontal lines would correspond to a strong sensitivity to changes in $NO_x$.

**2.3     Volatility Calculations**

Section 3.6 makes use of the vapour pressure (often expressed as a log value to the base of 10) to investigate the potential contributions to SOA. The UManSysProp facility was used to do this.(Topping et al., 2016) UManSysProp can estimate the vapour pressure of compounds represented as SMILES strings via a range of different group contribution methods. (Barley and Mcfiggans, 2010; O'meara et al., 2014) This work used predictions at 298 K throughout and used the 'evaporation'

technique, though sensitivity to all of the available prediction methods is described in Section 3.6.

### 3     Results and Discussion

### 3.1     Model Validation

As a test of the ability of the steady-state models to represent conditions present under ambient scenarios, the model results were compared to measurements collected in the summer of 2017 in Beijing. (Shi et al., 2019; Hamilton et al., 2021;

Whalley et al., 2021; Reeves et al., 2021; Newland et al., 2021; Mayhew et al., 2022) The $NO_x$ mixing ratios measured in the afternoon periods in Beijing ranged between 5 ppb and 20 ppb, and $O_3$ mixing ratios ranged from around 60 ppb to 140 ppb. Isoprene mixing ratios ranged up to around 2 ppb in the afternoon period, hence a typical value of 1 ppb was chosen for the steady state models. The concentration isopleths for inorganic species zoomed in to this representative range of $O_3$ and $NO_x$ mixing ratios are provided in the supplementary information (Section S1).

Measurements of OH reactivity ($k_{OH}$) during the afternoon period were between around 10 s$^{-1}$ and 30 s$^{-1}$ (Whalley et al., 2021), which the models reproduced at the appropriate $NO_x$ and $O_3$ mixing ratios by design due to the additional methane included in the model run for this purpose (Figure 4a). The modelled $NO_3$ reactivity ($k_{NO3}$) is around 0.4-1.9 s$^{-1}$ compared to the estimated value of around 0.5 s$^{-1}$ presented in Hamilton *et al.* (Figure 4b). (Hamilton et al., 2021). $k_{OH}$ and $k_{NO3}$ values at a wider range of $NO_x$ and $O_3$ mixing ratios is provided in Figure S1

In the $NO_x$-$O_3$ space corresponding to typical Beijing afternoon conditions, the models show NO mixing ratios of around 0.3-2.8 ppb (Figure 5a, Figure S2a), consistent with the low-NO observations in the afternoon period in Beijing with observed mixing ratios of around 0.25 ppb and 3 ppb. (Newland et al., 2021) The models show $NO_3$ mixing ratios of 0.4 to 2 ppt (Figure 6a, Figure S3a), which is slightly below the measured $NO_3$ mixing ratio in the afternoon of around 2 ppt. (Hamilton et al., 2021) The modelled OH concentrations are between $2.5 \times 10^6$ and $6.5 \times 10^6$ molecules cm$^{-3}$ (Figure 6b, Figure

S3b), which is slightly below the measured concentrations of around $7.5 \times 10^6$ molecules cm$^{-3}$. $HO_2$ is also reasonably predicted with a range between $4.2 \times 10^8$ and $9.1 \times 10^8$ molecules cm$^{-3}$, compared to measurements of around $2.5 \times 10^8$ molecules cm$^{-3}$ (Figure 6c, Figure S3c). (Whalley et al., 2021)



A series of sensitivity tests were carried out in order to assess the sensitivity of our conclusions to changes in model parameters. Four different parameters were adjusted: the concentration of isoprene, the concentration of methane, the dilution rate, and the time of day. These sensitivity tests were found to have little impact on the conclusions drawn in this work, and any potential impacts are discussed where required. Further details on tests is provided in the supplementary information (Section S2).

### 3.2 Mononitrates

The IHN concentration isopleth (Figure 7a) shows a strong similarity to the OH isopleth (Figure 6b), highlighting its rapid formation from the OH oxidation of isoprene and subsequent $RO_2 + NO$ reaction to form the nitrate group (Figure 1). This means that at high $O_3$ mixing ratios, IHN shows a strong dependence on $NO_x$, and the dependence on $O_3$ becomes more significant at lower $O_3$ mixing ratios.

IPN shows increasing concentrations with increasing $O_3$ and decreasing $NO_x$ (Figure 7b). This reflects the requirement for low NO concentrations to be present for two reasons. Firstly, high $NO_3$ concentrations are required to form the $INO_2$ radical by the reaction of isoprene with $NO_3$. Secondly, the $RO_2+HO_2$ reaction is required to form the hydroperoxide group of IPN, and so lower NO concentrations will reduce competition with the rapid $RO_2 + NO$ reaction. Additionally, $HO_2$ concentrations are highest under low $NO_x$ conditions (Figure 6c), further favouring the $RO_2 + HO_2$ reaction.

In contrast to the other nitrated species investigated here, ICN shows two peaks in concentration, one at very low $O_3$ and the other at very high $O_3$ (Figure 7c). This is because ICN can be formed from many different routes. Firstly, the abstraction of an H atom from IHN which provides a formation route under lower $O_3$ conditions, when OH and IHN are both high in concentration. Alternatively, under higher $O_3$ conditions, ICN can form from the reaction of OH and IPN, or the decomposition of nitrated alkoxy radicals (INO). The result of these multiple peaks is that under moderate $NO_x$ and $O_3$ conditions, the concentration of ICN is relatively insensitive to changes in both $NO_x$ and $O_3$. It is also important to note that the absolute concentrations of ICN predicted to form in these models are very low (the peaks in concentration corresponding to mixing ratios of just over 1 ppt), due to low production rates, which is consistent with low daytime ICN mixing ratios previously identified in Beijing. (Reeves et al., 2021; Mayhew et al., 2022)

Recent work has highlighted species with the formula $C_4H_7NO_5$ as potentially major oxidation products of isoprene. (Tsiligiannis et al., 2022) Consistent with previous modelling studies (Mayhew et al., 2022), these models largely form $C_4H_7NO_5$ from OH-initiated oxidation, meaning the concentration isopleth is similar to that of OH and IHN.

### 3.3 Nitrated Epoxides

Both INHE and INPE show a similar pattern as IPN in the $NO_x$-$O_3$ isopleths (Figure 8a,b), with the highest concentrations occurring at low $NO_x$ and very high $O_3$. INPE is reliant on high $NO_3$ and $HO_2$ in a similar manner to IPN as the $RO_2+HO_2$ step is still required to form INPE. While $HO_2$ is not required to form INHE via the alkoxy-epoxidation pathway, the formation of INHE from IPN is the major formation route under lower NOx conditions. Additionally, the alkoxy-epoxidation pathway to the formation of INHE relies on an $RO_2$-$RO_2$ cross-reaction. This cross reaction will be favoured under low-$NO_x$-high-$O_3$ conditions, where NO concentrations will be the lowest, meaning that the competition with the rapid $RO_2$-NO





reaction is minimal. This requirement for very high $O_3$ and low $NO_x$ means that we should expect very low concentrations of daytime INHE and INPE under typical urban conditions. This is consistent with modelling of Beijing which showed that while INHE may comprise a large fraction of night-time $C_5H_9NO_5$ compounds, the daytime contribution is very small.
(Mayhew et al., 2022)

In contrast, INCE shows reasonably high concentrations under high-$O_3$ mixing ratios at a range of $NO_x$ mixing ratios. The profile of these concentrations is very similar to the $NO_3$ isopleth (Figure 6a), and stems from the main formation route to INCE requiring the $NO_3$ oxidation of isoprene followed by an $RO_2 + NO$ reaction step. This formation route shares similarities with the dinitrates discussed in Section 3.4.

### 3.4   Dinitrates

As previously noted for INCE, Figure 9 shows that the steady-state concentrations of isoprene dinitrate (IDN) and isoprene dinitrooxyepoxide (IDNE) are very similar to the $NO_3$ concentration isopleth (Figure 6a). This is indicative of the formation route of IDN and IDNE, where an initial $NO_3$ oxidation is followed by the reaction of the resulting $RO_2$ with NO to form the second nitrate group (Figure 1 and Figure 3). At any daytime concentration of $NO_x$ where sufficient $NO_3$ is present to
perform the initial oxidation step, there will also be sufficient NO present to rapidly react with the resulting $RO_2$. Although the high $O_3$ mixing ratios observed in Beijing result in low NO concentrations compared to typical daytime concentrations in a polluted megacity, there is still ample NO present to react with peroxy radicals produced by the initial $NO_3$ oxidation. Figure 9 shows that at each $O_3$ there is a critical $NO_x$ concentration, above which the concentration of IDN, IDNE, and INCE is almost exclusively controlled by $O_3$ concentrations.  This critical $NO_x$ mixing ratio is reasonably low compared to typical
urban $NO_x$ mixing ratios, indicating that the concentration of dinitrates in urban environments may be largely controlled by the $O_3$ mixing ratios present.

As discussed in Section S2, increasing the total VOC in the model results in a broadening in the $NO_x$ axis of the transition between $NO_x$ and $O_3$ sensitive regimes for $NO_3$, and so the same applies to IDN, IDNE, and INCE. For a fixed ozone concentration, increasing the $NO_x$ from 0 will increase IDN, IDNE, or INCE concentrations due to the increased $NO_3$
resulting from the increased availability of $NO_2$ (Figure 10). Then, once the threshold $NO_x$ concentration is reached and $NO_3$ concentrations are not limited by the availability of $NO_x$, the concentration becomes controlled by $O_3$ and $HO_2$ (Section S2) creating the "hump" in concentration which then stabilises as $HO_2$ concentrations begin to decrease at high $NO_x$ and the $NO_2/NO$ ratio becomes increasingly controlled by the fixed $O_3$ concentration. This means that with high VOC concentrations, at a given $O_3$ concentration, reductions in $NO_x$ will result in increased IDN, IDNE, or ICNE concentrations
sooner than under lower VOC conditions.

### 3.5   Total Organonitrates
By summing the model concentrations for all organonitrates present in the mechanism, an isopleth of total organonitrates was obtained (Figure 11). This shows a band of high organonitrate concentrations at moderate $NO_x$. At high $O_3$, further changes to $O_3$ have little effect on the total organonitrate concentration. Total organonitrates also become less sensitive to



changes in $NO_x$ in this high-$O_3$ region. This band is the result of organonitrates produced by OH and $NO_3$ oxidation of isoprene. At low $O_3$ mixing ratios, the total organonitrates are dominated by OH-initiated species such as IHN (Figure 7a). Conversely, at high $O_3$ mixing ratios, $NO_3$-initiated species such as IDN comprise a larger fraction of total organonitrates (Figure 9a). This is illustrated in the pie charts in Figure 11 which show the organonitrate composition under different $NO_x$-$O_3$ regimes.

The composition breakdown also reveals that total isoprene organonitrates are dominated by IHN under most conditions, but this fraction decreases as ozone mixing ratios increase. At higher $O_3$, a large fraction of the composition comes from IDN, IDNE, and INCE due to their higher concentrations under high-$O_3$ conditions (Section 3.3 and Section 3.4). CH3NO3 comprises a substantial fraction of total organonitrates in these models. CH3NO3 can be formed by the OH oxidation of methane via the methylperoxy radical, and so its concentrations in these models are exaggerated due to the large amounts of

methane added to the model. Formation from the OH oxidation of methane comprises the majority of all of the methylperoxy formation in all of the models, excluding formation from the reversible decomposition of methane peroxynitrate (CH3O2NO2) which is balanced by the opposing formation reaction. CH2NO2OOH is also listed in Figure 11. CH2NO2OOH is formed from the products of isoprene ozonolysis, explaining the higher concentrations under high-$O_3$ conditions. There are no chemical losses in the mechanism for CH2NO2OOH, which likely explains the high contribution to

total organonitrates. The remaining portion of "other" organonitrates corresponds to a wide range of species, none of which contribute more than 6% to the total organonitrate sum in any models.

As noted in Section S2, changing the VOC concentration effects the position in $NO_x$-$O_3$ space where the maximum organonitrate concentrations are observed. In the case of total organonitrates, decreasing the total VOC concentration results in the band of high concentrations moving to lower $NO_x$. Figure S7 and Figure S8 show that with lower methane

concentrations in the model, the peak organonitrate concentrations occur at around 12 ppb of $NO_x$ whereas this increases to around 25 ppb of $NO_x$ in the high methane case. The peak organonitrate concentrations at each of these $NO_x$ concentrations are similar in each of these sets of model runs.

### 3.6 Volatility Assessment

One of the major motivations for studying isoprene nitrates is their potential to contribute to secondary organic aerosol

(SOA) by condensation or reactive uptake to existing particles. While the highest concentration species such as IHN and IDN may be most significant when considering the role of isoprene nitrates as $NO_x$ reservoirs, low concentration species can be much more important for SOA formation if they are of a sufficiently low volatility. As an estimation of the impact of changing $NO_x$ and $O_3$ on particle-phase processes, the log of the vapour pressure was estimated for each organonitrate in the mechanism based on the species' structure using UManSysProp.(Topping et al., 2016) As a measure of a compound's

volatility, a lower vapour pressure value corresponds to a less volatile compound which will more readily partition into existing particles. This volatility-based approach does not account for potential reactive uptake which is likely to be important for the epoxide species previously discussed.



Figure 12 shows the total organonitrate plot normalised to the vapour pressure value for each compound, which gives an estimation of the contribution of organonitrate uptake to SOA at each $NO_x$ and $O_3$ mixing ratio. Since the predicted vapour

pressures range over 10 orders of magnitude, lower volatility species can have a large effect on SOA formation despite their much lower concentration. The lowest vapour pressures predicted here are for the two MCM species NC524NO3 and NC524OOH. These two compounds comprise almost 100% of the normalised concentration in Figure 12 under all $NO_x$ and $O_3$ conditions, and their individual concentration profiles can be seen in Figure 13. The concentration isopleths for the 15 lowest volatility compounds are shown in Figure S17. Many of these species show profiles similar to that of NC524OOH,

with the highest concentrations occurring at low urban $NO_x$ concentrations.

According to Figure 12, reductions in $NO_x$ from typical urban conditions would result in higher normalised concentrations of organonitrates, meaning the contribution of isoprene nitrates to SOA may increase with decreasing $NO_x$ until very low urban $NO_x$ conditions are met. However, it is important to note the difficulty in representing the lowest volatility species in the isoprene oxidation mechanism. Many of these species are the product of multiple oxidation steps with large uncertainties

surrounding their rates of formation. Additionally, many of the lowest volatility species do not contain any chemical losses in the mechanism due to a lack of information on their reactions.

The low predicted volatilities of NC524NO3 and NC524OOH are the result of the many functional groups present in the molecules. Similarly oxidised species could be described as highly oxidised molecules (HOMs). (Bianchi et al., 2019) The chemistry of HOMs is not currently well represented in many mechanisms due to their varied autoxidation formation

pathways. For example, the profile of NC524OOH concentrations in Figure 13b results from the requirement of $HO_2$ to form the hydroperoxide group, hence the profile is similar to that of $HO_2$. If an $RO_2$ H-shift formation pathway to NC524OOH, or similar HOMs, were included in the mechanism, then this might alter the profile in Figure 12. (Vereecken and Nozière, 2020) It should be expected that the formation of HOMs would be sensitive to changes in ozone as the lower NO concentrations at higher ozone will reduce the competition of the $RO_2+NO$ pathway with the $RO_2$ autoxidation reactions

that form HOMs. Inclusion of improved autoxidation chemistry in the mechanisms would also increase the number of low volatility compounds produced from the oxidation of isoprene.

Figure S16 shows that the results presented here are reasonably insensitive to the choice of vapour pressure and boiling point prediction methods selected within UManSysProp. Section S3 also outlines the results using an alternative volatility estimation method, making use of only the molecular formula of each compound.

**3.7   Application to Less Polluted Environments**

In order to test the investigation of less polluted environments, a series of models were run at a range of lower $NO_x$ and $O_3$ mixing ratios. Measurements collected in the amazon rainforest were taken as an example of an unpolluted environment, and the model was adjusted to match typical isoprene mixing ratios and OH reactivity observed in this environment. The results from these models are presented in the supplementary information (Section S5).

The reduced $NO_x$ and $O_3$ mixing ratios used in these models mean that, despite the changes to isoprene and methane concentrations, they correspond well to higher resolution models of the low $NO_x$ and $O_3$ portions of the Beijing models.





Figure S19 shows that the total organonitrates in the Amazon models are sensitive to changes in $NO_x$, with increasing ozone slightly reducing total organonitrates.

As discussed in the introduction, IEPOX is one of the major precursors of isoprene SOA, particularly under low-$NO_x$

conditions where the IEPOX precursor, ISOPOOH, can form from oxidation by OH and further reaction with $HO_2$. This dependence on $HO_2$ means that the amazon models predict increasing concentrations of IEPOX as $NO_x$ is increased from close to 0, regardless of the $O_3$ concentration (Figure 14b). This is in agreement with findings from Shrivastava et al. who found increases in isoprene SOA resulting from increases in $NO_x$ and $O_3$ from an urban plume (Shrivastava et al., 2019) The increased SOA could be further explained by the increases in nitrated epoxides and dinitrated species predicted on increasing

both $NO_x$ and $O_3$ (Figure S19).

Methyl vinyl ketone (MVK) and methacrolein (MACR) are also often of interest in isoprene oxidation, particularly in pristine environments such as the Amazon where the production of MVK and MACR relies on the presence of NO.(Langford et al., 2022) This is illustrated in Figure 14d, whereas Figure 14c illustrates that the abundance of NO under typical urban conditions means that the MVK+MACR concentrations in the Beijing models are dependent on OH

concentrations.

## 4 Conclusions

The work presented here illustrates that each isoprene nitrate species will have a different $NO_x$-$O_3$ regime in which maximum concentrations will be produced. For example, the facile formation of IHN from OH oxidation means that daytime concentrations are largely dictated by the concentration of OH. Alternatively, the concentrations of species such as IDN,

IDNE, and ICN are largely dictated by the available daytime $NO_3$ as the reaction of $RO_2$ with NO is very rapid, even under low urban $NO_x$ conditions. Finally, IPN, INHE, and INPE only show the highest concentrations under low-$NO_x$-high-$O_3$ conditions due to their increased formation under high $NO_3$ concentrations and their requirement for low-$NO_x$ to avoid competition with the $RO_2$+NO reaction pathway.

The fact that the concentrations of different organonitrates will respond differently to changes in $NO_x$ and $O_3$ will have

implications for those considering pathways to reducing the concentrations of organonitrates in the atmosphere. The work presented here indicates that reductions in $NO_x$ may not reduce total organonitrate concentrations until low urban $NO_x$ conditions are met and that, for many of the species resulting from the daytime $NO_3$ oxidation of isoprene, organonitrate concentrations may be much more sensitive to changes in $O_3$ than in $NO_x$. Additionally, accounting for the volatility of the organonitrates can have very large impacts, and the models presented here are dominated by a small number of low volatility

compounds. An improved representation of late-stage oxidation and autooxidation is likely to improve the ability to predict the effect of changing $O_3$ and $NO_x$ on SOA formation.

As efforts are made to reduce $NO_x$, VOC, and $O_3$ concentrations around the world, care should be taken to ensure that the non-linearity of responses to changes does not result in unintended increases in important SOA precursors. For example, previous work has indicated that Beijing occupies a VOC-limited regime with respect to $O_3$ formation and that decreasing

$NO_x$ concentrations without a concurrent decrease in VOC concentrations would result in increased $O_3$. (Wei et al., 2019; Ren et al., 2021) Mapping this trajectory of decreasing $NO_x$ and increasing $O_3$ onto Figure 12 would suggest that SOA from isoprene organonitrates may increase in concentration due to the gradient with changing $NO_x$. However, the effect could be worsened when considering the reactive uptake of INHE, IDNE, INPE, and INCE which all favour formation under high-$O_3$ conditions. This may be further compounded or mitigated with changing VOC concentrations due to the impact on the

organonitrate isopleths as well as the non-linear behaviour of $O_3$ with changing $NO_x$ and VOC.

## 5    Competing interests

The authors declare that they have no conflict of interest.

## 6    Acknowledgements

This project was undertaken on the Viking Cluster, which is a high performance compute facility provided by the University
of York. We are grateful for computational support from the University of York High Performance Computing service, Viking and the Research Computing team.

## 7    Financial Support

This work was supported by the Leeds-York-Hull Natural Environment Research Council (NERC) Doctoral Training Partnership (DTP) Panorama under grant NE/S007458/1.

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





**Figure 1. Formation routes to form IHN, ICN, IPN, and IDN from the OH- and NO₃-initiated oxidation of isoprene. Additional isomers and reaction pathways have been omitted for clarity.**

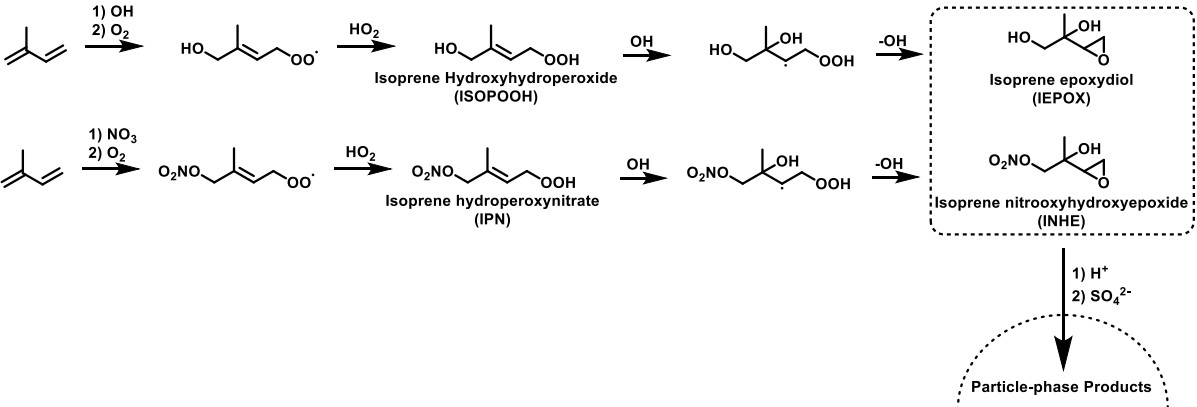

**Figure 2. Established formation routes to form IEPOX and INHE via the OH oxidation of stable hydroperoxide intermediates. Additional isomers and reaction pathways have been omitted for clarity.**



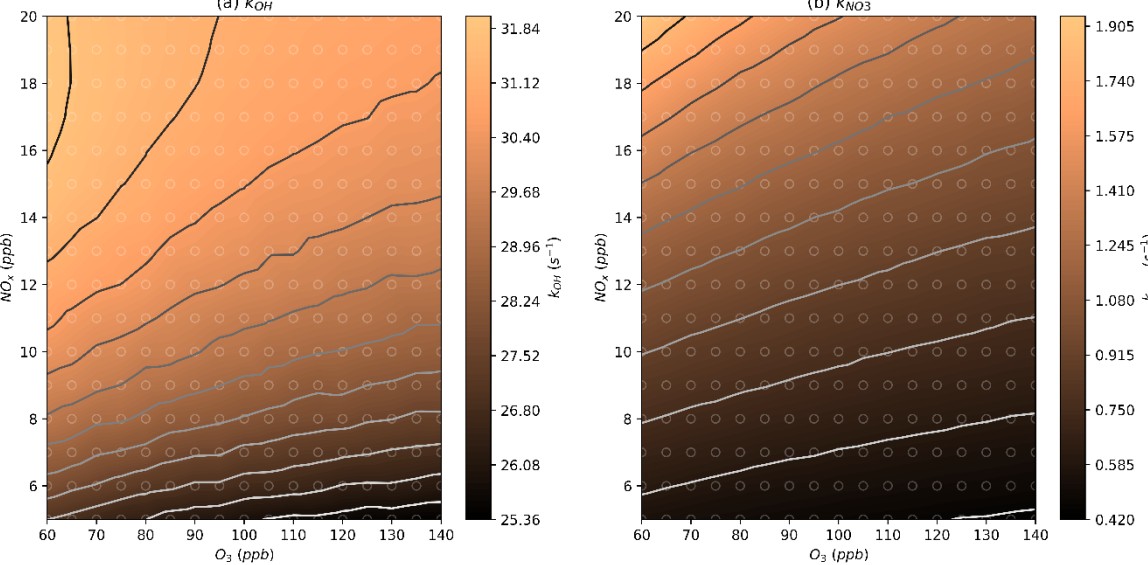

**Figure 3. The alkoxy-epoxidation pathway to form a range of nitrated epoxides, as proposed by Vereecken et al. 2021. Additional isomers and reaction pathways have been omitted for clarity.**

**Figure 4. Modelled steady-state kOH and kNO₃ values at different NOₓ and O₃ mixing ratios. Further details on interpreting these plots is given in Section 2.2.**





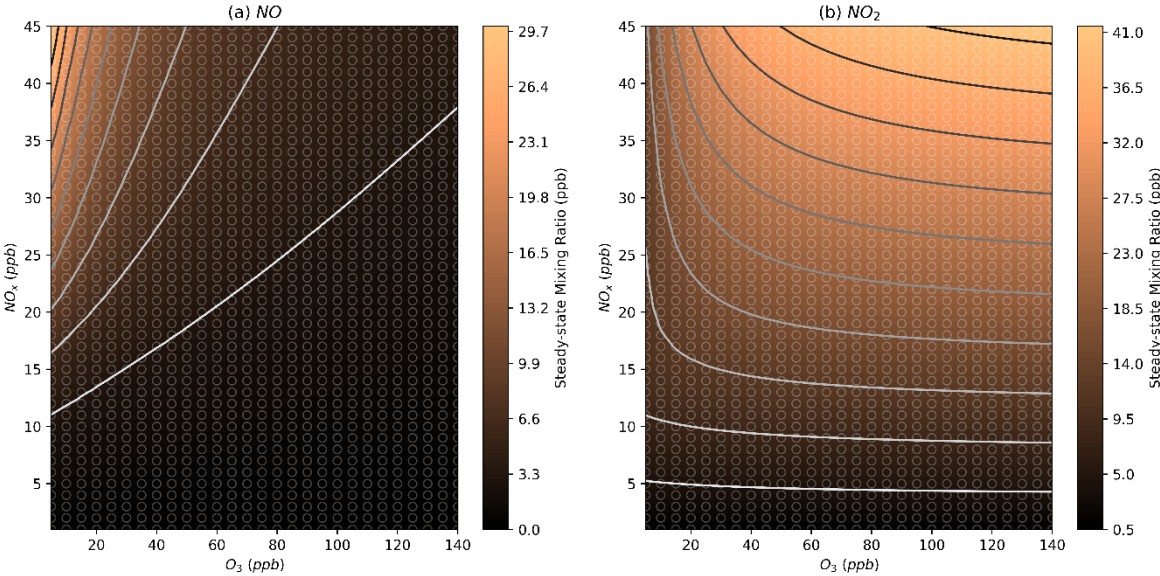

**Figure 5. Modelled steady-state mixing ratios of NO and NO₂ at different NOₓ and O₃ mixing ratios. Further details on interpreting these plots is given in Section 2.2.**

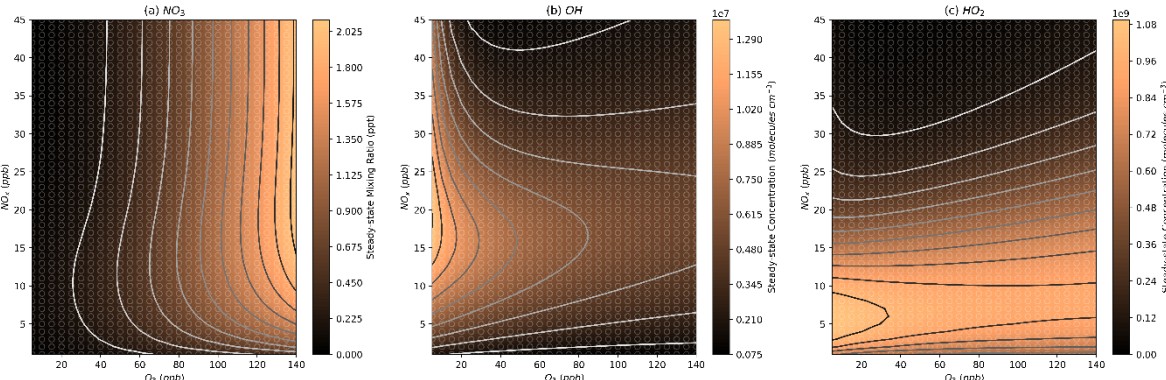

**Figure 6. Modelled steady-state concentrations of NO₃, OH, and HO₂ at different NOₓ and O₃ mixing ratios. Further details on interpreting these plots is given in Section 2.2.**





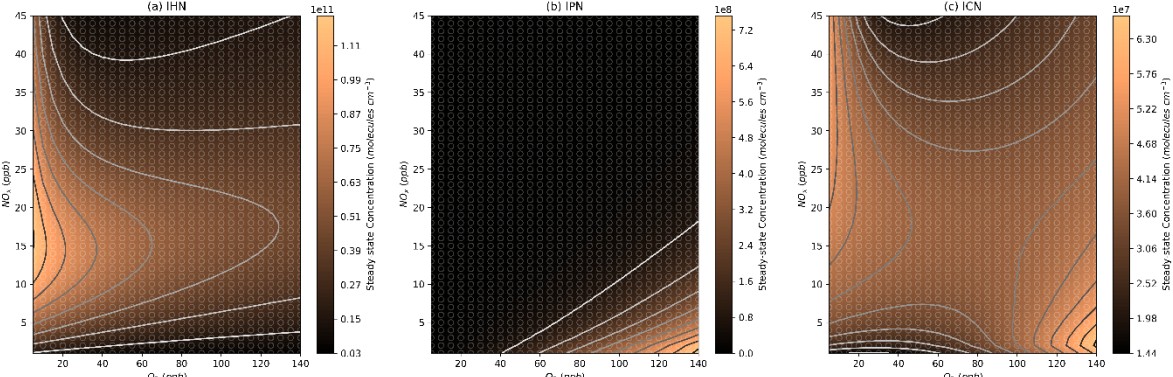

**Figure 7. Modelled steady-state concentrations of isoprene hydroxynitrate (IHN), isoprene carbonylnitrate (ICN), and isoprene hydroperoxynitrate (IPN) at different NOₓ and O₃ mixing ratios. Further details on interpreting these plots is given in Section 2.2.**

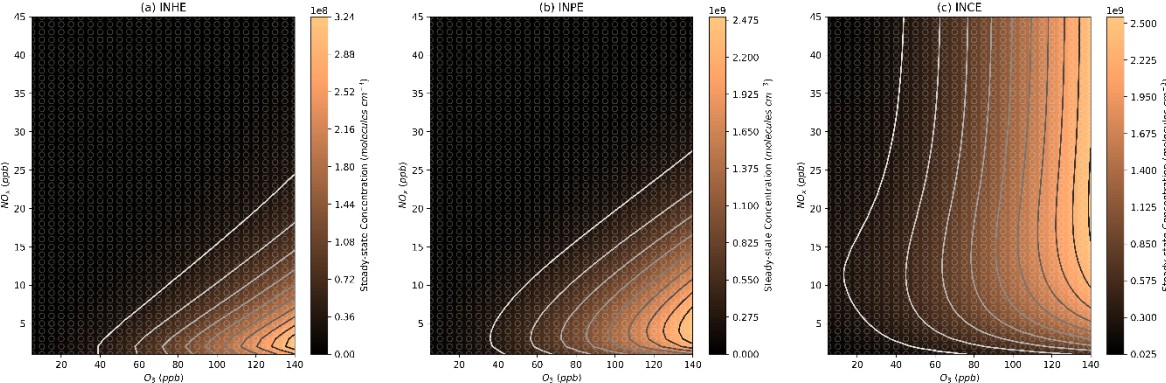

505 **Figure 8. Modelled steady-state concentrations of isoprene hydroxynitrooxyepoxide (INHE), isoprene carbonylnitrooxyepoxide (INCE), and isoprene hydroperoxynitrooxyepoxide (INPE) at different NOₓ and O₃ mixing ratios. Further details on interpreting these plots is given in Section 2.2.**



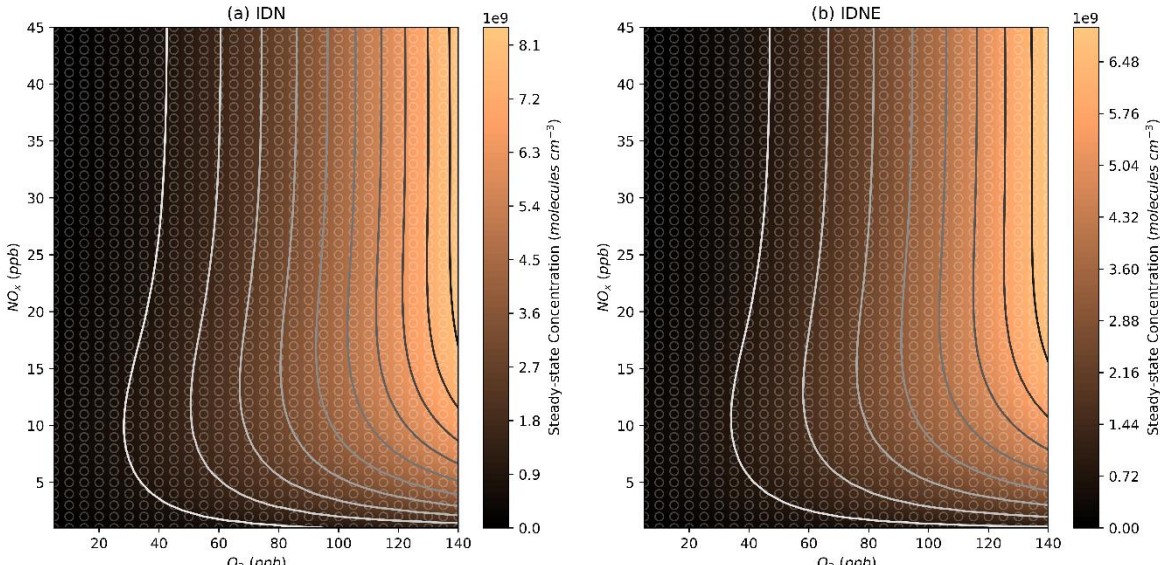

**Figure 9. Modelled steady-state concentrations of isoprene dinitrate (IDN) and isoprene dinitrooxyepoxide (IDNE) at different NO$_x$ and O$_3$ mixing ratios. Further details on interpreting these plots is given in Section 2.2.**

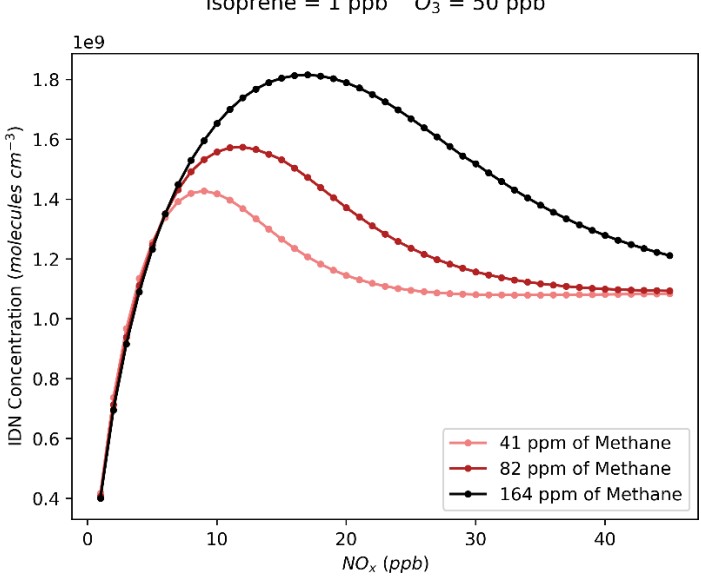

**Figure 10. Modelled steady-state concentrations of IDN with changing NO$_x$ in models runs with 50 ppb of O$_3$ and different mixing ratios of NO$_x$ and additional methane.**



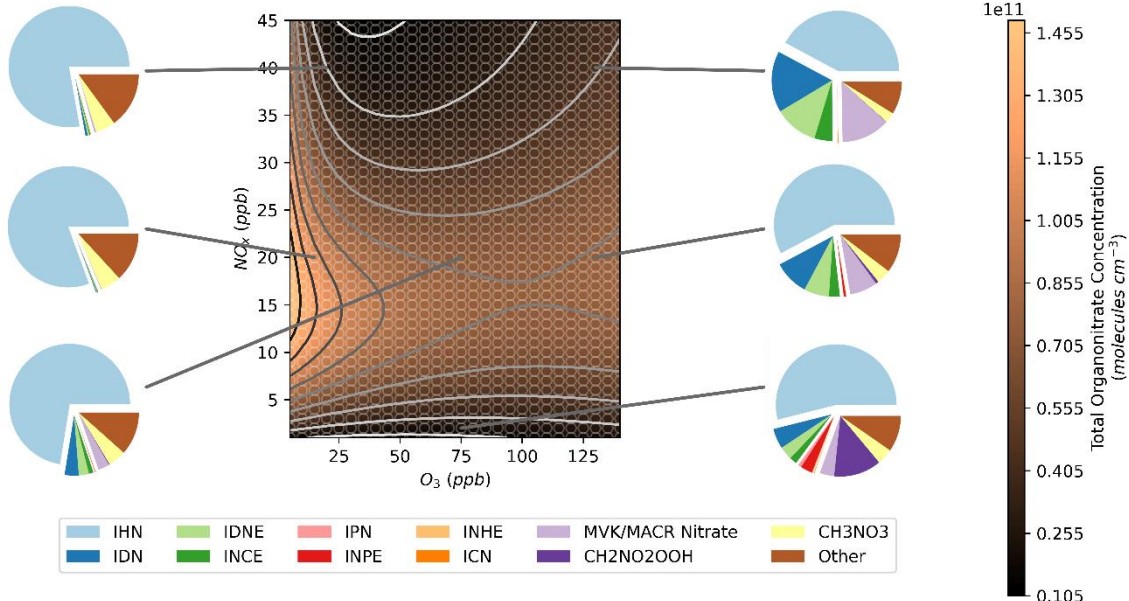

**Figure 11. Modelled steady-state concentrations of the total organonitrates at different NO$_x$ and O$_3$ mixing ratios along with the composition of the total organonitrates at selected NO$_x$ and O$_3$ mixing ratios. Further details on interpreting these plots is given in Section 2.2.**



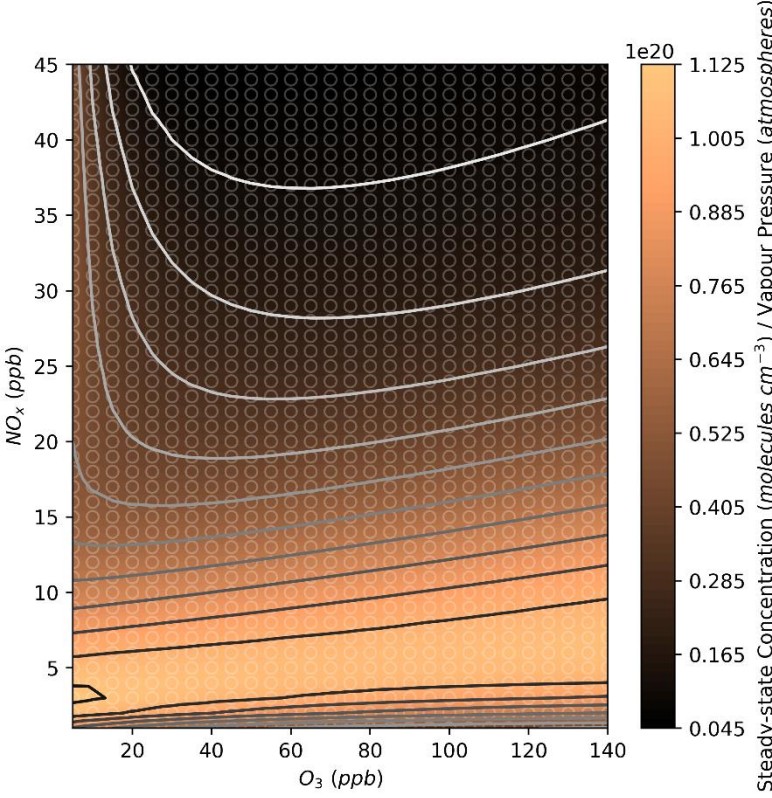

**Figure 12. Modelled steady-state concentrations of the total organonitrates normalised to each compound's estimated vapour pressure at different NOₓ and O₃ mixing ratios. Further details on interpreting these plots is given in Section 2.2.**





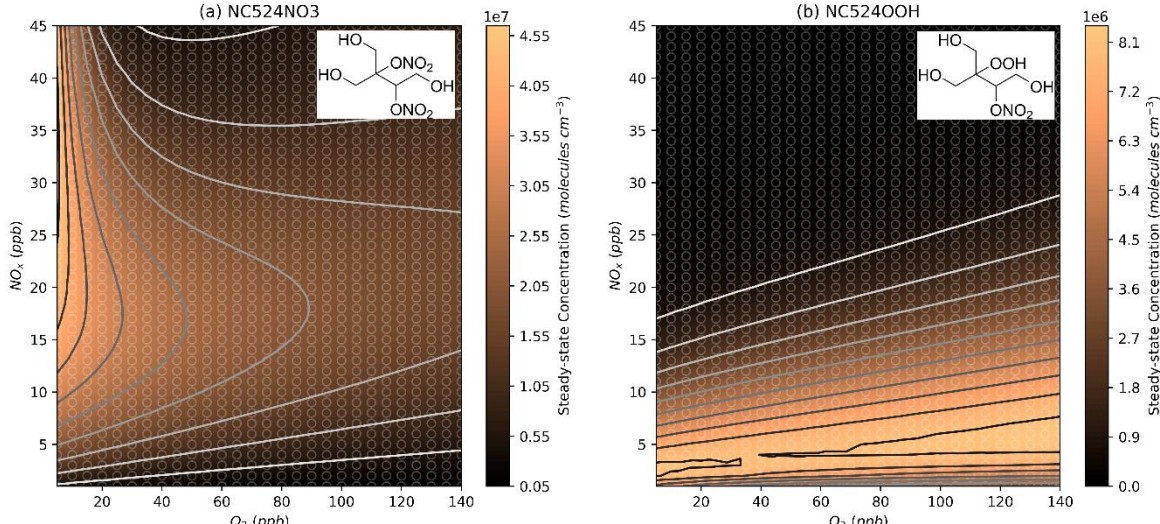

**Figure 13. Modelled steady-state concentrations of the MCM species NC524NO3 and NC524OOH at different NO$_x$ and O$_3$ mixing ratios. Further details on interpreting these plots is given in Section 2.2.**





**Figure 14. Modelled steady state IEPOX and MVK + MACR concentrations for the Beijing (a,c) and Amazon (b,d) models.**