# Peer review of "Daytime Isoprene Nitrates Under Changing NOx and O3"

_EGUsphere, 2023_

## Author Comment (AC1)

The authors are grateful for the reviewers' suggestions and contributions, particularly those that probed the applicability of this modelling approach to real-world scenarios. Below are the author's responses to each of the reviewer's comments. The responses to reviewers 1 and 2 have been numbered 1a-o and 2a-b, respectively.

**Reviewer 1**

This is a well-written and interesting theoretical modeling study showing how organic nitrates formed from isoprene oxidation during the daytime in Beijing, China are impacted by changing $O_3$ and $NO_x$ concentrations. The mechanism used in this work is state of the art and understanding how organic nitrates from isoprene form in the atmosphere especially under the highly polluted environment of Beijing, China is important. The results are quite interesting from a theoretical perspective, but sometimes hard to interpret from an atmospherically relevant perspective. Generally, due to the simplified nature of the modeling here, there are limitations to the conclusions that can be drawn and these limitations and the uncertainty they add to the conclusions should be more clearly specified in the text prior to publication as described more below.

Detailed suggestions:

Line 67: Can you include the exact Vereecken et al. reference here with the date since there are 2 references? Is the mechanism exactly the same as the one referenced in Vereecken? If not, can you provide details on any updates and/or a version number if available?

> 1a) The year has been added to the reference. One change to the mechanism that was previously not noted was a renaming of IHN species to integrate those formed from OH and $NO_3$ chemistry. In the FZJ mechanism as taken from Vereecken *et al.* 2021, ISOPCNO3 and EISOP1N4OH both exist in the mechanism (the first taken from the MCM and the second added through the revised $NO_3$ chemistry). These species both have the same structure, so in order to prevent this duplication, ISOPCNO3 was renamed to EISOP1N4OH. This has been explained in line 72: *"In order to make the species naming consistent between the MCM portion and the added chemistry of the FZJ mechanism, ISOPCNO3 was renamed to EISOP1N4OH as both identical species are present in the original FZJ mechanism."*

Line 73: Can you include a reference for the Beijing 2017 campaign?

> 1b) A reference has been included to Shi *et al.* 2019, which introduces the APHH Beijing campaign.

Line 81: Can you include a reference for why this dilution lifetime of 12 hours was selected?

> 1c) Previous modelling work makes use of a 24-hour dilution lifetime, however often alongside a separate deposition rate which may be calculated from a deposition velocity. Since the physical loss added here was largely to speed up the time taken to reach steady-state in the model, particularly for species without any losses in the mechanism, these physical processes were combined to a shorter lifetime of 12

hours. The sensitivity tests presented in our previous manuscript demonstrate that the conclusions are robust to changes in this selected value. The following sentence has been added at line 85: *"This value was selected based on a combination of the physical loss processes included in previous modelling work, and the impact of this decision is assessed in Section 3.1. (Mayhew et al., 2022; Edwards et al., 2014; Edwards et al., 2013)"*

Line 82, Why was this temperature and relative humidity selected? Are these the averages from the Beijing 2017 campaign at 16:00 local time too?

1d) These values were selected as standard temperature and pressure values, but the authors believe that the selected values are representative of the Beijing conditions. We do not have access to pressure measurements throughout the campaign, however the mean temperature between 16:00 and 17:00 was 304.4 K.

Section 3.1: Demonstrating that the ranges of various species (kOH, NO, NO2, NO3, OH, HO2, IHN, IPN, ICN, etc.) in the NOx/O3 space modeled matches with the observations is a good first step in building confidence in the model. However, higher confidence in your modeling approach would be to compare the exact observations for each NOx/O3 value at 16:00 during the Beijing 2017 campaign to that from the model. For example, you could use the same color bar but add stars or squares to represent the observations. Or just add plots in the supplement that are side by side using the same color bar, but one for the model and one for the observations. This would be useful to understand how well your model is replicating the contours and variation specifically rather than just confirming it produces results within the range of the observations. It would also show which space on these $O_3$ and $NO_x$ plots Beijing is typically in. The ranges of O3 and NOx may be exactly as you state, but the full O3 to NOx plots you are representing are not likely to be atmospherically relevant. For example, there is likely more of a curved line of actual space on these plots that reflects real conditions in Beijing, which would be useful for the reader to understand too. All of this would give the reader more confidence that the conclusions from your model on how reductions of $NO_x$ or $O_3$ will impact SOA and organic nitrates from isoprene oxidation are accurate. Doing something similar for the model results on for the Amazon would also be useful too.

1e) The purpose of these model validation steps was to verify that the modelled concentrations were approximately representative of real-world concentrations, rather than being able to directly reproduce the observations from Beijing. Further analysis has been done to produce the suggested plots, however the authors feel that their inclusion does not change the confidence in the modelling approach due to the limitations of this modelling approach for directly predicting real-world observations (e.g. fixed $O_3$, $NO_x$, VOC, and photolysis conditions). As such, the resulting plots are discussed here but not included in the revised manuscript, particularly in light of other comments by this reviewer that highlight the caution that should be taken when applying the results of these models to the real atmosphere.

The isopleth plots for Beijing displayed below have been produced in the same way as the original isopleths, but have the ambient Beijing data overlaid as stars, making use of the same colour scale as the underlying contour. These plots are also zoomed in to a smaller $NO_x$-$O_3$ region to allow the measured data to be seen clearly. The Beijing data was averaged between 15:30 and 16:30 and indexed by the average measured $NO_x$ and $O_3$ concentration at each 4pm period throughout the campaign. No plots have been produced for the Amazon data as the authors do not have access to the same detailed dataset for this campaign.

The first observation is the large scatter of the observations in $NO_x$, $O_3$ space. This is relevant to later comments from this reviewer with regards to the ability to move in straight lines around $NO_x$-$O_3$ space due to the non-linear interactions between NOx, $O_3$, and VOCs. This will be discussed later, in response 1k.

Generally, NO, $NO_2$, and $NO_3$ all show good agreement with the models, due to their concentrations largely being controlled by $NO_x$ and $O_3$, which are constrained. The OH measurements do not fit the contour produced by the models, however the values are of a similar magnitude to the modelled values. This is to be expected as the measurement points represent the whole suite of conditions present at 4pm during the Beijing campaign, whereas the models have constant VOC concentrations and photolysis conditions corresponding to average Beijing conditions. Finally, the $HO_2$ isopleth highlights the over prediction in $HO_2$ concentrations made by the models. This over prediction is noted in the original manuscript, though this plot highlights the extent of the over prediction. This is consistent with findings by Whalley et al. 2021 and Mayhew et al. 2022 which both note the over prediction in $HO_2$, particularly in the afternoon period under low-NO conditions. The authors have tested a model in which a sink of $HO_2$ is added to the mechanism in order to reduce $HO_2$ concentrations, however this results in an under-prediction in OH due to the $HO_x$ removed from the system.

The description of $HO_2$ in the manuscript at line 148 has been edited to ensure that the $HO_2$ over prediction is properly highlighted and explained: *"$HO_2$ is generally over predicted with a range between $4.2 \times 10^8$ and $9.1 \times 10^8$ molecules $cm^{-3}$, compared to measurements of around $2.5 \times 10^8$ molecules $cm^{-3}$ (Figure 6c, Figure S3c). (Whalley et al., 2021) This is consistent with previous modelling studies that indicate an over-prediction of $HO_2$ by models, particularly under the low-NO afternoon conditions being investigated here. (Mayhew et al., 2022; Whalley et al., 2021) Furthermore, models which included an additional sink of $HO_2$ to bring it in line with measurements resulted in an under-prediction of OH due to the $HO_x$ removed from the system."*.

**NO**

[Figure]

**NO2**

[Figure]

**OH**

[Figure]

**HO2**

[Figure]

[Figure]

In Figure 7, 8 & beyond, why plot these organic nitrates in molecules cm^-3 instead of mixing ratio like you did for NO, NO2, and NO3 in Figures 5 and 6? It seems easier for the reader to interpret if you put these in mixing ratios as is typically done even if you need to switch from ppb to ppt for the lower yielding organic nitrates. This seems important so that viewers understand your results because IHN has such higher concentrations than the others and this is hidden and easily missed by readers by a very small 1eX written above the color bar for each plot.

> 1f) The figures were initially displayed in molecules cm$^{-3}$ since a direct comparison between model and measured concentrations is not appropriate. In the case of NO, NO2, and NO3, this comparison was useful for model validation but the focus of the organonitrate isopleths should be on the change in organonitrate concentrations under changing NO$_x$ and O$_3$ (i.e. the shape of the contours). However, the authors accept that presenting the data as mixing ratio values may be more useful for comparisons between isopleths, so the units have been changed on all of the relevant plots in the main paper and SI.

Section 3.2 & 3.3: From the authors previous work, Mayhew et al., 2022, the diurnal cycle of these isoprene organic nitrates particularly those from NO3 oxidation have complicated diurnal cycles. Can you comment on whether only showing the steady state at 16:00 impacts the interpretation of your results for atmospherically relevant conditions? How important are IPN, IHN, ICN, the nitrated epoxides (INHE, INPE, INCE) that formed the previous night and then linger into the day and potentially form other oxidation products for the total organic nitrates at 16:00? These organic nitrates formed during the nighttime and

their oxidation products are not considered in your current modeling and past work has suggested nighttime formation of organic nitrates to be an important contribution of total organic nitrates (e.g., Kenagy et al., 2020, https://doi.org/10.1029/2020GL087860). How does this assumption to not include the nighttime formation of these organic nitrates impact your conclusions?

1g) The authors fully agree that the night-time formation of organonitrates is important for a full understanding of the atmosphere in $NO_x$-impacted environments and that the simplifications that come with this modelling approach do not account for this formation route. However, while studies like Kenagy et al do demonstrate the importance of night-time organonitrates for the morning period, the diurnal profiles presented in Mayhew et al. 2022 show that once production ceases, the concentrations of isoprene organonitrates formed overnight (IPN and ICN) rapidly decrease throughout the early morning so that concentrations would be very low by 4pm. It is difficult to assess the impacts of the downstream reaction products of these night-time organonitrates, however, the attempts to reproduce typical $k_{OH}$ and $k_{NO3}$ values should account for this chemistry to some extent.

The final paragraph of Section 2.1 has been edited to outline this model simplification and now reads as follows: *"These models are designed for comparison between one another to gain insight into the impact of changes in $NO_x$ and $O_3$ on organonitrate concentration. The models show the concentrations at steady-state for the provided photolysis conditions, chemistry, and dilution rate, which is in contrast to the constantly changing photolysis and dilution encountered under ambient conditions. This means that chemistry that would occur at other times (e.g. night-time chemistry) does not contribute to the results of these models. Previous work has demonstrated the importance of night-time chemistry for understanding organonitrates, but also that many of the organonitrates produced overnight will decrease in concentrations over the early-morning period to reach low concentrations in the afternoon.(Kenagy et al., 2020; Mayhew et al., 2022) The downstream chemistry of these night-time organonitrates, and other night-time species, will have some impacts on day-time chemistry that are not captured by the models presented here. However, Section 3.1 illustrates that the conclusions made in this paper are applicable to the Beijing afternoon conditions being investigated and that the conclusions are robust to changes in the modelling approach."*

Section 3.5 and Figure 11: See comment above, how would not including the contribution of organic nitrates formed during the night impact your results here? Potentially, the text here should be clearer to emphasize that this is not the expected fraction of organic nitrates at 16:00 in the real atmosphere, but instead only the fraction of organic nitrates formed directly from isoprene at 16:00 and does not include organic nitrates formed during the night or organic nitrates formed during the night and further oxidized during the day.

1h) As noted previously (response 1g), Mayhew *et al.* 2022 demonstrates that ICN and IPN (which are predominantly produced during the night) decrease to very low concentrations during the day. Mayhew et al 2022 also shows that while IHN can be formed during the day and night, the daytime IHN is dominated by OH-initiated

isomers. Additionally, since the total organonitrate composition is dominated by primary organonitrates, it is unlikely that an individual night-time organonitrate oxidation product would contribute significantly to the total organonitrate sum.

The same is not true for the volatility-weighted assessment, where low concentrations of highly oxidised compounds can have a large impact. So a statement has been added at line 277 to address this possibility: *"There is also the potential for the oxidation products of night-time species to be low volatility compounds that would contribute to SOA but would not be captured by these models which represent the steady-state concentration of organonitrates at 16:00, without the contribution of chemistry occurring at other times of the day."*

Line 232: You are also not considering the uptake of tertiary organic nitrates and hydrolysis that occurs in the atmosphere and this should also be mentioned here (Vasquez et al., 2020)

1i) A sentence has been added to note this in line 252: *"Furthermore, the hydrolysis of organonitrates, particularly tertiary nitrates, may reduce particle-phase concentrations. (Vasquez et al. 2020)"*.

Conclusions: See comment above for Section 3.2. Your conclusions here only represent organic nitrates formed during the day from isoprene directly and not organic nitrates formed during nighttime or organic nitrates formed during the nighttime and further oxidized during the daytime. This is important to emphasize as it is different from what will occur in the actual atmosphere. Some description of how this impacts your overall conclusions would be useful so that readers know how to interpret your results.

1j) In light of this comment and previous similar comments, the following changes have been made:

Line 306: *"The work presented here illustrates that each isoprene nitrate species will have a different $NO_x$-$O_3$ regime in which maximum concentrations will be produced during the afternoon period"*

Line 320: *"It is important to note that the simplified models presented here represent the formation of day-time organonitrates at 16:00 and will not capture the effect of organonitrates produced during the night-time or those produced from the oxidation of night-time products."*

Lines 289 to 293: If you think about the typical ozone isopleth as a function of NOx and VOC. If VOCs are constant as you are doing in this study (methane + isoprene) then each NOx level is going to produce a certain ozone concentration, so this would produce a curved line across each of these graphs where the conditions are atmospherically relevant. When you talk about some organic nitrates would benefit from reductions in NOx and others with reductions of ozone here and throughout the text, can you further explain what you mean by this under atmospherically relevant conditions? Generally, if you keep NOx the same, and you want to reduce ozone, you would have to reduce VOCs, but this is not what you are simulating here because you keep VOCs constant in all of these graphs (i.e., isoprene +

methane). Similarly, if you reduce NOx, you will impact ozone, so you are never going to go straight down the y-axis of one of these graphs in the real atmosphere. Can you explain more how the reader is supposed to interpret these plots from an atmospherically relevant perspective? How is ozone supposed to be reduced (i.e., moving left along the x-axis) when keeping VOCs and NOx constant in the real atmosphere especially as you are also holding all the other levers constant too (e.g., temperature, photolysis)?

1k) As previously noted (response 1e) when looking at the measured data mapped onto the model isopleths, the conditions at 4pm in Beijing do cover a wide range of $NO_x$ and $O_3$ concentrations, because of different VOC and photolysis conditions corresponding to each real-world point. However, it is correct that under a fixed VOC concentration, there will be areas of $NO_x$-$O_3$ space that are inaccessible. Ideally, these plots would include the third dimension of VOC concentration however this would soon become very difficult to visualise and interpret. This was the rationale behind inclusion of plots demonstrating the change in the isopleths under lower and higher isoprene and $CH_4$ concentrations, and the discussion of these changes in the text.

A note has been added to Section 2.2 (line 117) to emphasise the real-world $NO_x$-$O_3$ interactions that are important for interpreting these isopleth plots: *"Under real-world conditions, the $O_3$ concentrations will be determined by the non-linear interactions of $NO_x$ and VOCs, which is in contrast to the models where $NO_x$, $O_3$, and VOCs are all constrained. This means that some sections of the model isopleths will be inaccessible. For example, some amount of $O_3$ will form in the presence of $NO_x$ and VOCs, so occupying the upper-left corner of the isopleths may not be possible outside of the constrained model scenario. Similarly, it may not be possible to map real-world changes to one dimension (such as a decrease in $NO_x$) onto the isopleth plots without accounting for a change in the other dimension (such as a change in $O_3$)."*.

Ultimately, this paper is intended to highlight the non-linearity in the chemistry of daytime isoprene organonitrates, and the role of daytime $NO_3$ chemistry under high $O_3$ conditions, rather than presenting a prediction of expected changes to organonitrate concentrations that might result from changes to $NO_x$ or VOC emissions. The authors believe that the presented isopleth plots highlight these points despite some of the positions in $NO_x$-$O_3$ space being inaccessible and there being some uncertainty about what trajectories can be followed around the $NO_x$-$O_3$ space. When discussing the potential future changes to $NO_x$ and $O_3$ in Beijing, we do note the potential increase in $O_3$ occurring alongside $NO_x$ decreases and factor this into the discussion of changes to SOA precursors under future scenarios. However, some additional changes have been made throughout the manuscript to ensure that the conclusions of this work are correctly limited to a discussion of the sensitivity of isoprene organonitrate chemistry rather than presenting the results as expected changes that may result from changes to $NO_x$ or VOC emissions.

Line 52 onwards has been changed to read: *"This work describes efforts to investigate the sensitivity of daytime isoprene organonitrate chemistry to changes in $NO_x$ and $O_3$ concentrations through a series of steady-state models."*

Line 301 onwards has been changed to read: *"Figure 12 suggests that many of the lowest volatility daytime isoprene organonitrates may form in higher concentrations under higher $O_3$ and lower $NO_x$. Furthermore, INHE, IDNE, INPE, and INCE all favour formation under high-O3 conditions and may be subject to reactive uptake to the particle phase."*

For your conclusions on SOA, under atmospherically relevant conditions, the picture may be more complicated than you are implying here. As Pye et al., 2019 (https://doi.org/10.1073/pnas.1810774116) nicely describe as NOx emissions decline, so do oxidants, which ultimately leads to less VOC oxidation and thereby SOA production. How do the results of this study impact your conclusions here?

1l) Changes in oxidant concentrations resulting from changes to $NO_x$ (or $O_3$) will be captured by these models through the chemistry included in the mechanism, and this is a key factor for explaining many of the conclusions made in the manuscript. For example, one of the central themes of the paper is the interplay between OH- and $NO_3$-initiated organonitrates, and the different regions where OH and $NO_3$ are highest which impacts the areas where the highest organonitrate concentrations are found. The decline in oxidant concentrations at low $NO_x$ described in Pye *et al.* can also be seen in the model results presented here. For example, the OH concentration curve presented Figure 2b of Pye et al. is similar to a vertical transect taken through one of the OH isopleths in this work (e.g. Figure 6b in this work). Both show a gradual rise in OH as $NO/NO_x$ concentrations increase followed by a peak and gradual decrease at very high $NO/NO_x$. There does not seem to be any contradiction between the work presented here and the work presented in Pye et al., particularly since this work is solely focusses on daytime organonitrate production and Pye et al. is solely focusses on HOM formation.

However, one of the key messages regarding SOA production that the authors wish to convey in this conclusions section is that some of the conclusions around SOA formation are limited by the difficulty of representing low-concentration low-volatility species (such as the HOMs discussed in Pye et al.) in existing chemical mechanisms. The manuscript states that "An improved representation of late-stage oxidation and autooxidation is likely to improve the ability to predict the effect of changing $O_3$ and $NO_x$ on SOA formation".

You are correct in your statements that in Beijing concurrent reductions in VOCs and NOx are likely needed to reduce O3 since reductions in NOx only would likely lead to increases of ozone. On this note, can you provide further context on which regions of your plot in Figure 12 would represent the NOx transition line where below this level of NOx you can expect O3 to start reducing when NOx emissions are reduced? From your current NOx to O3 plots, it is difficult to know where this regime change is expected to occur and this seems important if you are using this plot to state SOA may increase as NOx emissions decline.

1m) As a response to previous comments by this reviewer, we have changed our discussion of the applications of this work to real-world emissions in an effort to emphasise the paper's focus on the sensitivity of the chemistry to changes in $NO_x$ or

O$_3$, rather than expected changes in pollutants such as SOA under changes to NO$_x$ emissions. For example, the change to line 301 in response 1k.

However, in response to this comment, we have also investigated the fate of RO$_2$ in the models in order to assess the relevant importance of RO$_2$+NO$_x$ reactions as opposed to RO$_2$+RO$_2$ or RO$_2$+HO$_2$ reactions. This gives an indication of whether the model is representing a NO$_x$-limited or radical-limited O$_3$ production regime. As the plots below indicate, RO$_2$+NO$_x$ reactions dominate the RO$_2$ fate until very low urban NO$_x$ concentrations of up to about 3ppb. This is consistent with the existing statements in the paper that Beijing occupies a VOC-limited regime and that decreases in NO$_x$ may increase O$_3$ concentrations in the city.

[Figure]

These plots have not been included in the revised manuscript due to the large number of figures already included in the paper, and the effort made in previous changes to move away from directly mapping future changes in NO$_x$ and O$_3$ onto the isopleths. However, the authors are willing to include the plots in the supplementary information at the editor's discretion.

Minor items

Figure 1: Typo in the Isoprene carbonyl nitrate should be ICN instead of IHN underneath it.

1o) This has been corrected.

**Reviewer 2**

This paper presents some calculations on the concentrations of isoprene oxidation products under a range of NOx and O3 conditions. The paper is detailed and a useful contribution to the literature.

I recommend two additions to give the reader who is interested in the big picture context a deeper understanding of what matters about these results.

First, there are quite a few studies of organonitrates more generally that place isoprene in context including several comprehensive reviews. Orienting the reader to this literature would help them to appreciate the ways in which this study answers an important and unanswered question (to the extent that it does).

2a) The Introduction section does make reference to several studies and reviews investigating both isoprene and organonitrates, however the authors acknowledge that there could be more of an emphasis on the questions that this work aims to answer and the contribution it provides to the understanding of isoprene organonitrate chemistry. The following text has been added:

Line 52: *"However, these $NO_x$ and $O_3$ dependencies have not previously been investigated."*.

Line 55: *"This work also aims to identify the role of $O_3$ concentrations in daytime $NO_3$ chemistry and determine the $NO_x$ and $O_3$ concentrations that facilitate this understudied reaction pathway."*.

Second, the potential for observational tests of the ideas described in this paper including both specific molecular observations of the isoprene daughters described and more general assessments with methods that observe total organo nitrates in he gas or particle phase would be helpful.

2b) A paragraph has been added to the end of the paper outlining the experimental observations that would help to validate the findings from this work and overcome the simplifications that are necessarily part of using a box model in this way. The text is as follows: *"Future chamber and field studies could validate the findings from this work by making more comprehensive observations of total and speciated organonitrates under different $NO_x$, $O_3$, and VOC concentrations. The comparative nature of these isopleth plots mean that the organonitrate measurement would not necessarily need to be calibrated, provided that the instrument response to specific organonitrates can be assumed to be constant. This makes long-term measurements made with iodide chemical ionisation mass spectrometry ($I^-$-CIMS) a promising dataset as $I^-$-CIMS is very sensitive to these multifunctional compounds, but calibration is often difficult. (Mayhew et al., 2022; Lee et al., 2014)"*

---

## Author Response (AR2)

Thanks for considering my suggestions. In particular, thanks for updating your text to more clearly describe the limitations to your conclusions to the real atmosphere from using a simplified modeling approach. This is now much clearer to the reader in all cases and there is less risk of the reader being misled on how to interpret these results. I suggest publication after consideration of the minor suggestions listed below:

Section 3.1: I disagree that comparing the model to observations directly does not add confidence to your model. I think you should include these observations in your paper, but ultimately you can make that call as the authors of the paper. My major concern that the simplified nature of the modeling here limits the conclusions that can be drawn to the real atmosphere does not mean you cannot compare to observations. You can still compare to observations while mentioning that there is expected to be some disagreement due to the simplified nature of the model. However, if there is general agreement, which for most of the species there is except for OH and HO2, then this lends confidence in the model and it shows where there might be limitations for example with HOx.

> The authors appreciate the value that may be added by these plots, however we have taken the decision not to include them to the final paper. The authors maintain that readers should not expect direct agreement between the measurements and models in this way. While the inclusion of the plots may serve to demonstrate some limitations of the modelling approach, the authors believe that these limitations have been stated elsewhere in the paper, particularly in light of the first round of reviews.

> For example, the OH measurements do not reproduce the model isopleth. However, this is to be expected as each day of the campaign will involve different atmospheric conditions, such as the amount of sunlight and the concentration of VOCs. The OH plot does show that the models are generally representative of an average OH concentration across each of the days, which is why the paper discusses model-measurement comparisons in terms of campaign averages and ranges.

> Regardless, these plots will still be publicly available to those interested in the intricacies of this modelling approach, through the public review process for this paper. We hope that this access to the plots is satisfactory.

Section 3.6: The hydrolysis of tertiary organic nitrates are not only going to impact your particle-phase concentrations, but the dominant isomer of the isoprene hydroxy nitrates is a tertiary organic nitrate and it's loss in the atmosphere in a polluted region like Beijing may be dominantly due to uptake to aerosols and so your organic nitrate gas-phase distribution is also going to be impacted by not including this process.

> A note has been added to Section 3.6 to highlight that these particle-phase hydrolysis processes could have knock-on effects on gas-phase organonitrate concentrations, particularly in the case of IHN.

> Line 254: "This particle-phase hydrolysis may then have knock-on effects for the gas-phase organonitrates, particularly where the tertiary nitrate isomer comprises a large fraction of the composition, such as for IHN."

Conclusions: I agree with the editor that since you cite a number of references in the introduction based on work from the CF3O- CIMS that have led to better understanding of isoprene organic

nitrates that this would also be a good tool for validating this work and should be mentioned here too.

The final line of the conclusions section has been made more general, to reference both I⁻-CIMS and CF$_3$O⁻-CIMS. I have also included a reference to Br⁻-CIMS which has also been used in the isoprene chamber experiments discussed in Carlsson *et al.* 2023.

Line 338: "This makes long-term measurements made with chemical ionisation mass spectrometry (CIMS) a promising dataset as various CIMS techniques using a range of reagent ions (including I⁻, Br⁻, and CF$_3$O⁻) have been shown to be very sensitive to these multifunctional compounds, but calibration is often difficult. (Mayhew et al., 2022; Lee et al., 2014; Schwantes et al., 2019; Carlsson et al., 2023)"